# Evaluation of oral health services and challenges faced by oral health practitioners working in Nyarugenge, Rwanda

**Emmanuel Nzabonimana**[1,2]* , **Yolanda Malele-Kolisa**[2], **Phumzile Hlongwa**[3]

**1** Department of Preventive and Community Dentistry, School of Dentistry, University of Rwanda, Kigali, Rwanda, **2** Department of Community Dentistry, School of Oral Health Sciences, Faculty of Health Sciences, University of the Witwatersrand, Johannesburg, South Africa, **3** Department of Orthodontics, School of Dentistry, Faculty of Health Sciences, University of Pretoria, Pretoria, South Africa

☯ These authors contributed equally to this work.
* enzabonimana@cartafrica.org

## Abstract

### Objective

Oral healthcare professionals encounter considerable challenges while providing services to curb the oral disease burden. The aim of this study was to explore the challenges faced by oral health practitioners providing oral health services in Nyarugenge, Rwanda and to appraise the availability and adequacy of oral hygiene equipment, instruments, and materials.

### Methods

This was a cross-sectional concurrent mixed methods study. The quantitative and qualitative parts were independent during data collection and analyses and merged during the interpretation phase. All seven public health facilities and 14 dental professionals working in Nyarugenge were included in the study. Data were collected using an audit checklist and an in-depth interview guide. Descriptive statistics were used to analyze the quantitative data. The interviews were analyzed using thematic content analysis.

### Results

Most of the dental health professionals were dental therapists (n = 11), women (n = 9), aged 31–40 years (n = 7), and with 11–20 years' experience as oral health practitioners (n = 6). There were five health centers and two hospitals that were audited for equipment, instruments, and materials. The audit of the facilities revealed that most facilities have dental equipment and instruments, but none have adequate preventive dental instruments and materials. Four broad themes emerged from the interviews with the oral health practitioners, namely human resources, supply chain management, patients' oral health awareness and service rendering, and strategic management and administration. The most significant challenges oral health practitioners faced included high dental practitioner/patient ratios, lack of adequate and appropriate equipment and materials, patients' lack of oral health awareness, and a lack of administrative support.

**Data Availability Statement:** All relevant data are within the manuscript and its Supporting Information files.

**Funding:** "EN was supported by the Consortium for Advanced Research Training in Africa (CARTA). CARTA is jointly led by the African Population and Health Research Center and the University of the Witwatersrand and funded by the Carnegie Corporation of New York (Grant No. G-19-57145), Sida (Grant No:54100113), Uppsala Monitoring Center, Norwegian Agency for Development Cooperation (Norad), and by the Wellcome Trust [reference no. 107768/Z/15/Z] and the UK Foreign, Commonwealth & Development Office, with support from the Developing Excellence in Leadership, Training and Science in Africa (DELTAS Africa) programme. The statements made and views expressed are solely the responsibility of the Fellow. The funders had no role in study design, data collection and analysis, decision to publish, or preparation of the manuscript.

**Competing interests:** The authors have declared that no competing interests exist.

## Conclusions

Well-established community preventive interventions, such as a mobile oral health App, could reduce the patient/provider ratio by increasing population awareness of oral health and encouraging healthy behaviours. The management of the health facilities should address the human resource challenges and equipment supply chain issues.

## Introduction

In their daily work, oral health professionals (OHPs) fight common oral diseases, especially periodontal disease and dental caries, which are some of the most common non-communicable diseases and are even more prevalent than cardiovascular disease [1]. However, in this fight, either through prevention or treatment, OHPs encounter many challenges. The first challenge identified by researchers is financial because dental services are expensive and considered among the most expensive healthcare treatments [2]. Consequently, dental services are difficult to afford, especially in low-income countries and even in higher-income countries among people with lower income, senior populations, and marginalized communities such as refugees, migrants, and people with disabilities [3, 4]. In the USA, more people complain of financial barriers to accessing dental care, irrespective of age, income level, and insurance type, than any other healthcare. Most treatments are not always covered by national insurance schemes, and they cover only a small range of services [5, 6]. When confronted with the problem of financing, health professionals tend to focus on clinical procedures that provide them with better renumeration instead of on risk assessment or prevention, and many prefer working in private dental clinics where they can earn better salaries [7]. This suggests that public institutions are frequented by individuals who may be unable to afford costly dental treatments [8]. Singh et al. stated that most dentists target profit compared to patients' dental problems [9]. Therefore, some dental services are not offered in public institutions due to payment challenges and the expensive nature of some dental procedures.

The provision of oral health services is further compromised by lack of infrastructure, equipment, instruments, and materials. The lack of infrastructure at primary health centers and community health centers is a major hindrance to oral health delivery in India, and this may be due to the lack of dentists that are appointed in government decision-making bodies [9]. In care homes for the elderly in Germany dental professionals have requested that a dental chair with a light should be available in every care home to facilitate dental treatment [10]. The lack of OHPs has also been reported as a problem in South Africa [7]. The lack of infrastructure leads to few working stations and few dental care providers, causing long patient waiting times, and the lack of equipment and materials impinges on the options and quality of treatments [11, 12]. In Nigeria, the lack of equipment and materials has been found to jeopardize the treatment options for patients and made OHPs to choose treatment procedures for which instruments and materials are available, such as tooth extraction [6]. The Rwanda National Strategic Plan for Oral Health 2019–2024 highlighted that in district hospitals and health centers, dental equipments, especially dental chairs, were either not available or not functioning due to lack of maintenance and lack of space for oral health services [13].

The biggest challenge experienced by OHPs when offering services has been linked to lack of human resources, which resulted to low dentist/patient ratios [14]. Different dental specialists are needed to collaborate in comprehensive oral examinations and treatment [15]. The problem of a shortage of dentists has been raised in the USA [16], and the situation is even worse in low- and middle-income countries. In Nigeria, the dentist/patient ratio was reported

to be 1:38 600, whilst in Tanzania it was 1:360 000 but in high-income countries, it was 1:2 000 [17]. Many health facilities in Nigeria have been found to have only one dental professional who does not have colleagues to consult about difficult cases [6]. OHPs' workload makes them overlook important steps in the diagnosis process, such as comprehensive oral examinations, comprehensive treatment care and oral health education. It is important to perform a comprehensive oral examination, including screening for early signs of all oral diseases, because it has been observed that most patients consult only for pain and dental emergencies [17, 18]. However, preventive measures are often neglected despite the goals of oral healthcare being a commitment to prevention [7, 9].

Apart from the financing of dental care, the availability of infrastructure and equipment, and the low dentist/patient ratio, the behavior of many patients has also been a challenge. The lack of oral health awareness among the population concerning early signs and symptoms of oral diseases, preventive measures, and regular dental check-ups resulted in late dental care-seeking behavior [6]. Even when patients have an infected tooth that can be conserved, it is challenging to convince them to comply with alternative treatments such as root canal treatments because they visit the dentist only when the pain is unbearable and insist that the tooth be removed [6, 19]. All these challenges are mostly due to oral healthcare not being included in the primary healthcare system. Oral healthcare should be integrated into primary healthcare to make it available to all people [6]. The adoption of healthy oral health preventive measures cannot take place if patients and caregivers are not given oral health education by OHPs on good oral hygiene practices such as frequent brushing, good brushing technique, interdental cleaning, regular dental check-ups, diet, and smoking cessation for the prevention of both periodontal diseases and dental caries [16].

The above-mentioned challenges can be addressed through good management. Some managerial barriers to good and equitable oral health identified in the literature are unclear goals, inadequate organization and management, poor planning, little emphasis on evaluation, and failure to plan for adequate manpower and use of auxiliaries [9]. It has been observed that dental service is often neglected during recruitment and procurement processes [6, 8, 13]. Literature demonstrates that there is international neglect and a lack of political prioritizing of oral health [16, 20]. Many public policymakers do not understand or value oral health as part of general health, making it a lower priority [9]. The affordability of dental services, the skills of OHPs, and the availability of dental equipment and materials all play important roles in the provision of comprehensive oral care, and this is the responsibility of the management of health facilities [17].

This study explored the challenges faced by OHPs in providing oral health services to the community of Nyarugenge, Rwanda. In addition, the availability and adequacy of oral hygiene equipment, instruments, and materials were also appraised. This study formed part of a larger doctoral study titled "Oral Health in Nyarugenge District of Rwanda: The Role of Mobile Applications in Oral Health Education".

## Methods

### Study design

This study was a cross-sectional concurrent mixed methods study that used both qualitative and quantitative methods. The qualitative part of the study aimed to explore the challenges faced by OHPs when providing oral health services to the community of Nyarugenge District, and the quantitative part was an audit of oral health services in Nyarugenge District. The quantitative and qualitative parts were independent during data collection and analyses but were merged during the interpretation phase.

## Setting

Rwanda is one of 56 countries in Africa. Rwanda is divided into four provinces and Kigali City. The four provinces each have 27 districts, and Kigali City has three districts, namely Gasabo, Kicukiro, and Nyarugenge [21, 22]. The study was conducted in seven public health facilities that offer dental services in Nyarugenge District. Five health facilities are public and two are managed conjointly by the government and the Roman Catholic Church. These health facilities service all patients, including those using community-based health insurance (CBHI).

## Study population and sampling strategy

The quantitative section of the study assessed the seven public dental health facilities in Nyarugenge District, and for the qualitative section, OHPs working in the seven dental health facilities were individually interviewed. No sampling was done. All seven dental health facilities in Nyarugenge District (n = 7) and all OHPs working in these facilities were included in the study. Fourteen OHPs were recruited for in-depth interviews.

## Data collection

The data collection tools for both the quantitative (observation checklist) and qualitative data collection (interview guide) were adapted from a study conducted in 2018 by Josephin et al. [23]. The data collection was started on 18 July 2022 and ended on 01 August 2022. The checklist items for the quantitative data collection assessed the availability, adequacy, and functionality of dental equipment. The in-depth interview guide had 12 questions related to various scenarios experienced by OHPs when providing oral health services to the community. The 12 questions asked during the in-depth interview were as follows: (1)How is your work now? (2) What suggestions can you make so that your work becomes much easier? (3)Do you manage to give oral health education to every patient that comes to you? (4)Are you able to provide scaling and polishing treatment to all the patients who need treatment? (5)Do you get time to sterilize and reuse instruments for scaling and polishing according to need? (6)Do you get time to give oral hygiene instructions to patients after a procedure? (7)Are you satisfied with the quality of treatment that you are offering to your patients? (8)What happens when one of the equipment in the dental clinic stops working or needs repair? (9)When dental materials like polishing paste finish, is it replaced immediately? (10)Do you feel that you are at risk while providing treatment? (11)In your opinion, what would decrease your workload? (12)What are your suggestions to improve oral hygiene treatment facilities in the dental clinic?

Following informed consent, the primary investigator (EN) conducted the interviews with the OHPs in a private room. The in-depth interviews were conducted in English and interpreted in Kinyarwanda, the local language, when necessary. Each interview lasted around 30 minutes. The responses of participants were audio recorded, transcribed verbatim, and saved as individual Microsoft Word documents soon after the interview. At the end of the interview, an audit of the dental equipment was performed using the observation checklist.

## Data analysis

Descriptive statistics were used to analyze the quantitative data using percentages and frequency distribution. The dichotomous responses, 1 equals *yes* and 0 equals *no*, to identify the availability, adequacy, and functionality of the dental equipment, instruments, or material needed for scaling and polishing were recorded. Data were entered manually on an Excel spreadsheet and then imported into Stata version 16 (manufactured by StataCorp LLC in

USA) for analysis. For the qualitative data, the recorded interviews were transcribed and imported into NVIVO software (manufactured by QSR International) for a detailed analysis.

The interviews were analyzed using thematic content analysis [24]. The first step in the analysis was to look at the participants' own words and phrases without preconceived notions or classifications. We examined in deep the responses of each participant's answer to all the questions.

To ensure trustworthiness, two additional researchers, an experienced qualitative researcher in health systems and a dental specialist, participated in the development of the codes by reading the participants' responses independently from the primary researcher to establish an inter-coder agreement. Once the initial analysis was completed, the team liaised to discuss the codes generated independently and to reach an agreement on the codes and themes.

### Ethical statement

The research received ethical approval (M220213) from the University of the Witwatersrand Human Research Ethics Committee (Medical). Additionally, permission was obtained from the relevant healthcare authorities, including the Rwanda Institutional Review Board (IRB) ethical committee (No. 234/CMHS IRB/2022) and the National Health Research Committee (No. NHRC/2022/PROT/26). The written informed consent form was obtained from the participants.

## Results

### Description of the study sample

Seven facilities were audited for the availability, adequacy, and functionality of equipment for preventative dental services. Furthermore, 14 OHP from these facilities were interviewed about the challenges they face when providing oral health services to the community.

### Frequency distribution of health facilities with essential dental equipment and instruments

Table 1 shows that five health facilities had dental chairs but only four of these chairs were fully functional. Only two health facilities had adequate dental chairs, dental lights, and dental

**Table 1. Preventive dental equipment, instruments and material audit availability.**

| Equipment/instruments | Availability | | Functionality | | Adequacy | |
|---|---|---|---|---|---|---|
| | Yes | No | Yes | No | Yes | No |
| Dental chair | 5 | 2 | 4 | 3 | 2 | 5 |
| Dental light | 5 | 2 | 4 | 3 | 2 | 5 |
| Dental stool | 5 | 2 | 5 | 2 | 2 | 5 |
| Dental compressor | 5 | 2 | 5 | 2 | 5 | 2 |
| Suction lines | 6 | 1 | 5 | 2 | 5 | 2 |
| Sterilizing unit | 7 | 0 | 7 | 0 | 7 | 0 |
| Ultrasonic scaler unit | 6 | 1 | 5 | 2 | 3 | 4 |
| Ultrasonic tips | 6 | 1 | 5 | 2 | 0 | 7 |
| Manual scaling set | 7 | 0 | 7 | 0 | 4 | 3 |
| Suction tips | 5 | 2 | 5 | 2 | 4 | 3 |
| Instruments for polishing | 5 | 2 | 5 | 2 | 4 | 3 |
| Prophylaxis handpiece | 5 | 2 | 5 | 2 | 4 | 3 |
| Prophylaxis angle | 5 | 2 | 5 | 2 | 4 | 3 |
| Prophylaxis cup/brush | 5 | 2 | 5 | 2 | 4 | 3 |

stools. All health facilities had functional and adequate sterilizing units. The manual scalers were available and functional in all seven health facilities, and five facilities had available and functional equipment and instruments for scaling and polishing. However, the adequacy was low concerning the ultrasonic unit (n = 3) and ultrasonic scaler tips were inadequate. Functional prophylaxis handpieces, prophylaxis contra-angle, and prophylaxis cups/brushes were found in four of the facilities. Polishing material was also inadequate in three facilities.

**Frequency distribution of health facilities with essential dental material.** The materials for oral hygiene instructions, oral examination instruments and personal protective equipment (PPE) availability and adequacy from the facilities were analysed. Only two health facilities out of seven had a demonstration model, toothbrush, and dental floss for interdental cleaning during oral hygiene instruction. None of the health facilities used a mobile application for oral health education. All seven health facilities had dental probes, mouth mirrors, and dental trays, but only six had periodontal probes. However, all these instruments were adequate in only four health facilities. Six of the health facilities had all personal protective equipment and they were adequate. The mouth masks were available and adequate in all the health facilities.

**Demographic characteristics of OHPs.** Overall, the total number of OHPs was 14, with five based in health centers, three in district hospitals, and six in referral hospitals. Most respondents were dental therapists (n = 11), and the dental surgeon (n = 3). The females (n = 9) were predominant, and the majority of the participants were aged between 31–40 years (n = 7). The 41–50 age group included 4 OHPs, and in the 21–30 age group had 3 OHPs. The years of practice of the OHPs revealed that (n = 6) OHPs had more than 10 years, 6–10 and 1–5 years of experience had 4 OHPs respectively.

*Emerged themes*. Although overlapping, four broad themes emerged from the responses of OHPs in the in-depth interviews, shown in Table 2, and these themes are discussed in the following subsections.

## Human resources

Apart from the OHPs working at the district hospital who reported that the patient/practitioner ratio was manageable, all the other OHPs complained about patient overload. In five of the health centers, there was only one dental therapist managing all the patients. The understaffing resulted in complaint-oriented service instead of comprehensive holistic dental management. Furthermore, the OHPs had to rush to clear the long queues and cannot not perform other treatments such as scaling and polishing or dental fillings since many patients have acute pain. In addition, ensuring good infection control can be challenging when the practitioners

**Table 2. Themes that emerged from OHPs' experiences in Nyarugenge District.**

| Theme | Description |
|---|---|
| 1. Human resource | • Poor OHP/patient ratios<br>• Lack of dental assistants |
| 2. Supply chain management | • Insufficient of equipment<br>• Insufficient instruments,<br>• Insufficient dental materials |
| 3. Patient education and service rendering | • Lack of public awareness<br>• Delayed oral healthcare seeking practices<br>• OHP workloads |
| 4. Strategic management and administration | • Health system responsiveness<br>• OHPs' attitudes and behavior (pleasant, rude, or caring attitude);<br>• Management of service provision; inadequate or lack of communication |

are working alone without a dental assistant. The following comments by the OHPs gave a clearer picture of the situation:

> "The number of patients is higher compared to the number of dental practitioners. Sometimes you have to rush in order to clear the line of patients." (Participant 2)

> "When there are many, we sometimes don't do a deep intraoral examination and concentrate only on the chief complaint instead of doing a comprehensive oral exam." (Participant 5)

> "Sometimes I am not happy due to the large number of patients. So I cannot satisfy all their needs. We can skip doing some treatments like scaling and dental fillings due to that number." (Participant 12)

> "You understand that this is a big challenge. If we have to work as four hands and you are there alone, helping yourself, it is not correct." (Participant 11)

Another challenge expressed by the OHPs working at the district hospital and the referral hospital was the lack of specialists. The waiting list and time are long for patients who must be seen by a specialist, and it was reported that some patients with oral cancer have died while waiting for treatment. Furthermore, since there is no dental surgeon at the district hospital, even patients requiring simple endodontic treatments and impacted teeth removal are referred to the hospital. Unfortunately, when patients arrive at the referral hospital, they can be turned back because of lack of dental material, and then they must get another referral to the Rwanda Military Hospital. That was distressing for both the patients and the practitioners, as shown in the following comments:

> "Bring also dental surgeons so that more services would be offered from here. Patients would receive a bigger package of dental care forms here; references would be reduced." (Participant 3)

> "I think they should train more specialists. Currently, we have only one maxillofacial surgeon dealing with accidents and emergencies. You find that a patient with orofacial cancer can die without even being examined by the specialist." (Participant 7)

It is imperative to recruit more OHPs for primary oral healthcare services because if this is not done, the quality of care offered in public health facilities will continue to be compromised.

## Supply chain management

The OHPs reported that they do not have sufficient equipment, instruments, or materials to offer dental care to the community. The audit findings from the checklists showed that two health centers did not have a dental chair, and of the five health centers that did have a dental chair, only four had a fully functional one. At the referral hospital, only two chairs out of four were available and could be used for ultrasonic scaling. The situation was similar at the district hospital, where only one chair out of two was available, fully functional, and could be used for ultrasonic scaling. Three health centers had one dental chair each, but only two of these chairs could be used to perform ultrasonic scaling. In general, no dental clinic had adequate equipment for the number of patients. As shown in the following comments, the OHPs expressed that the lack of equipment, instruments, and materials impacts the oral healthcare they provide to the community:

"Concerning equipment, since we have many patients who need our services, it would be much better if we had more equipment. Currently, we have only one dental chair. It would be better if we had two of them. More equipment should be purchased to serve the population better." (Participant 8)

"Most of the time we don't have enough materials for all the patients. Some patients get good service while the others get lower service because some materials have finished." (Participant 9)

Competing priorities in hospital settings between dental services and other services, such as maternity or pediatric services, were cited by OHPs as reasons for the lack of dental resources. The dental equipment could be damaged and not be repaired, and consumables were finished and not procured, resulting in patients waiting a long time before receiving treatment. This is a challenge at almost all the surveyed health facilities. Usually, in public health facilities, tender and procurement processes take a long time, which means that some items that are received are used immediately because the waiting time was so long. The following comments by participants express this challenge:

"That is also a challenge. It doesn't mean that the materials are nowhere to be found, only that the process is long . . . sometimes you get them too late when it is no longer possible to help the patients. You can make a request today and wait a whole year for delivery." (Participant 9)

"The priority is given to the maternity, pediatric, and neonatology services. The dental service lags in the process of supplying.. . . we always wonder if the dental service is futile or is not generating money so that its revenues cannot help to buy a scaler. If a scaler has been functioning for like two years, it is not understandable that the money generated during that period cannot buy another one to replace the damaged one. That is always a challenge." (Participant 1)

Even though six of the seven dental clinics had ultrasonic scaler units, most of them did not have enough scaler tips, which led to overuse until the tips were no longer efficient. In many of the facilities, they can book only two patients per day, despite the need to book more patients. The availability and functionality of the scaling instruments were optimal, but when compared to the number of patients who needed this treatment, it was inadequate. The problem is worsened by the fact that dental clinics are no longer allowed to have a sterilizer on the premises, which would have allowed them to sterilize scaling instruments between patients. The new protocol means sterilization can only be done at the main sterilization unit of the hospital or the health center. These new protocols result in delays in the sterilization process and the availability of sterile instruments, negatively impacting services. The OHPs described the problem as follows:

"These types of equipment and materials should be in enough because many patients are having a lot of calculus. Sometimes even when they have another chief complaint, you might wish first to clean their teeth before proceeding with the filling or the extraction in order to work in a safe oral environment." (Participant 3)

"I had requested 30 scaler tips, but I have only two. But even those two are no longer functioning perfectly because they are overused." (Participant 11)

"I become stressed because I see patients who need (preventive) services. Telling someone to come back after one month is very frustrating. You would like to serve them immediately but due to the shortage of equipment, you are obliged to give a rendezvous one month later." (Participant 11)

"We sterilize in a big container from the autoclave at the theatre and in the morning, they bring them back and put them in small boxes." (Participant 5)

After scaling, it is necessary to polish dental surfaces to prevent a rapid re-accumulation of calculus. However, only three sites had the necessary polishing paste, and most of it was of poor quality and could not perform good polishing. The service of polishing teeth posed another challenge because it is not covered by the CBHI scheme, which is mostly used in public health facilities. Therefore, many patients attending public health facilities cannot afford to pay for teeth polishing services. The result is unpolished dental surfaces that attract dental plaque and calculus more quickly, which discourages patients and makes them believe scaling is more harmful than beneficial to their oral health. The following comments by the participants explain this problem:

"We would also like to get the polishing paste whenever we need to use it. People start to be aware and to look for better oral hygiene." (Participant 3)

"The request of consumables is done on time, but since we are in a public hospital, the process of delivery is very long so at given times we stop doing some procedures because dental materials are not there. That process causes us to experience stock-outs." (Participant 2)

"The only consumables they care to avail are those linked to tooth extraction. But for the others, the administration is reluctant." (Participant 12)

The adequacy of diagnostic dental instruments was low in all seven health facilities considering the number of patients received daily, except for personal protective equipment. The COVID-19 pandemic improved the availability and adequacy of personal protective equipment in all health facilities because administrators of health facilities realized the importance of personal protective equipment in protecting patients, practitioners, and colleagues.

## Patient education and service rendering

The heavy workload of OHPs leads to them overlooking important practices such as oral health education. The OHPs said that it is almost impossible to give oral health education to every patient they examine. The OHPs give superficial oral health education to patients with halitosis or poor oral health status, but only OHPs working in two health facilities said they found time for thorough one-on-one oral health education. This is possible because another dental therapist was recently recruited at one health center, which means the district hospital staff are no longer overloaded. This resulted in improved OHP/patient ratios. This proves that the will to do oral health education is there but that the practitioner/patient ratio makes it impossible. However, the OHPs confirmed they give postoperative instructions to every patient after treatment regardless of their overload. The following comments by the OHPs show the extent of the problem:

"It is not possible to give oral health education to every patient. Most of the time, we do it for patients with poor oral health status or with halitosis. That is when we tell them that they have to brush their teeth." (Participant 5)

"We only tell them to remember to brush their teeth in the morning and at night before going to bed. Only that." (Participant 7)

"Giving oral health education to every patient is challenging because patients are many. You only tell them about the act itself you are about to do for them or, after completing the procedure, giving them related instructions. Otherwise, it is very challenging unless you do mass education, but they don't come at the same time." (Participant 4)

"There are also some people who don't want to hear what you are telling them, thinking that you are wasting their time while you wanted them to acquire some awareness." (Participant 3)

Patients are unaware that it is better to prevent oral diseases than to cure them and that teeth can be cleaned professionally. They often realize that there are some deposits on their teeth and try to remove them with charcoal, which causes more dental problems like enamel abrasion. Most of the patients seek dental services because of pain. Even when OHPs see the need for scaling and polishing and gave them an appointment, the patients do not come for the treatment. According to the OHPs, most patients do not listen to the advice they are given during individual or mass oral health education and consider it a waste of their time. One OHP said that the reason for this is that there are no OHPs recruited in the community to teach them oral hygiene measures and to screen their mouths for early detection of diseases. Community oral health prevention should be established and consolidated because it will tackle the problem of oral health at the root, especially since individual dental self-care by tooth brushing, including interdental cleaning, is the most important preventive measure for periodontal disease and dental caries. Furthermore, patients should cooperate with OHPs and quit harmful habits such as smoking, alcohol, and drug abuse.

"There should be dental personnel affiliated in the community to educate people about oral health. That would be very good". (Participant 4)

Oral health education is also a challenge because of a lack of teaching aids and promotional material. Only one health center, managed by sisters of the Roman Catholic Church, had all the necessary didactic materials for oral health education. The facility had a big teaching model and a big brush they could use to clearly show all the parts of the oral cavity and how to clean them. Participant 8, who works at this center, said, "First of all, we have didactic materials in our office. We have a big toothbrush and a big model that we use to show them the technique for tooth brushing. We also show them the gum and the tongue on the model. You see that this model is showing all the parts of the mouth". The only other health facility with a small dental model teaching aide is the district hospital. At all the other centers, oral hygiene measures are taught using gestures, and it is impossible to verify whether patients understand what they are taught. The OHPs did not even have dental floss to show patients how to clean interdental spaces and had to teach patients only in theory. The lack of time due to patient overload and the lack of didactic materials negatively affect oral health education, which is the basis for oral disease prevention. Participant 5 explained it as follows:

"We don't have typodonts and materials which would help us to explain the technique for dental brushing. When we do it only in theory, we are not sure of the outcome. Didactic materials should be available on our consultation tables, but we don't have them . . . we only give oral health education in theory, but we cannot verify if they understood well or not." (Participant 10)

The large number of patients seeking dental care reflect the oral health status of the community. The poor oral health status of the community could be improved if oral health education is improved through measures such as active, practical education through people's mobile phones. All the interviewed OHPs were amazed by such an idea and recognized that the application would make their job much easier because patients would come to them with some knowledge about oral health and ask questions to clarify the information they received on their phones. If the information is updated regularly with clear audio-visual materials, it will positively impact people's oral health, even in remote areas. The following comments show the OHPs reactions to the idea of a dental mobile application:

"People would be aware of when and how to perform oral hygiene and consequences of poor oral hygiene." (Participant 14)

"That is where that application would play a role in raising the awareness of our clients about oral health. The communication between dental patients and dental practitioners would be smoother. That wouldn't take us a lot of time trying to convince them about the best treatment." (Participant 1)

"That is why the application you were telling me about would be very useful because problems start at the community level, not at the health facility . . . since oral problems can be prevented, and the application can help them in that, it would be good if it was installed in people's phones and was used. It would remind them how to clean the mouth." (Participant 4)

An oral health education mobile application would promote community oral health awareness, help children, youth, adults, and even old people maintain their oral health and show them early signs of disease so they seek dental care in time. It would inform the population of the importance of regular dental check-ups instead of waiting to get dental care only when in pain. It would also improve people's oral health literacy, which would ease their communication with OHPs and increase their compliance with preventive measures.

## Strategic management and administration

Many of the challenges faced by the OHPs could be addressed by management. Management plays a crucial role in staff recruitment, motivation and the procurement of proper dental equipment and materials. Management's role is also related to reducing patients' waiting time by increasing the number of working stations by supplying dental chairs and employing dental assistants. A good workplace should be a safe environment for both patients and practitioners, and OHPs should feel valued by the administration.

"It is challenging to serve such a big number of patients while we don't have enough working stations so that each one might work from his/her dental chair". (Participant 9)

Poor strategic management and administration were particularly observed at two health centers where OHPs lacked basic equipment such as dental chairs and basic instruments despite constant requests. At one health center, dental extractions and manual dental scaling are performed using archaic equipment. The OHP working there reported that requests sent to the Ministry of Health can take long and that they often receive no response.

"Here, we don't have dental equipment. We have asked for them, but we haven't yet received them . . .. I would be happy if they bought for me new and modern equipment". (Participant 6)

   

At another health center, the dental chair had been taken during the COVID-19 pandemic and given to another health center in a different district because the health center had become an isolation center for COVID-19 patients. The participant explained that they have asked the administration to return the equipment but to no avail:

> "If the dental chair and all needed instruments, equipment, and materials were available so that when the patient comes and needs any treatment they might get it, this would satisfy me". (Participant 10)

The OHPs are frustrated because their requests for dental instruments and materials are ignored by management. The OHPs believe that the lack of dental representatives in the procurement committees is the reason their orders are not approved. Procurement and supply chain committees/personnel are unfamiliar with dental jargon, instruments, consumables, and requirements and do not consult the OHPs when they require information. The OHPs explained the problem as follows:

> "They don't supply us with dental materials on time so that we might serve our patients in a good and satisfying way." (Participant 3)

> "Another thing, even when they provide, usually they bring a different item from the one you requested. As a recent example, we requested drilling burs and they brought polishing burs, while we needed inverted cone bars." (Participant 2)

Only health facilities managed conjointly with the Roman Catholic Church seem to prioritize dental services and procure what was needed, as shown by the following comments:

> "There is no problem here even for equipment. I can say that our administration is valuing dental service even more than I do." (Participant 8)

> "Concerning sterilization of instruments, we never lack instruments here. We are privileged because this health center belongs to the Catholic Church so basic instruments are there. Even though they cannot be enough for all the patients, we clean and sterilize them. Our instruments are safe because we have a sterilizer in which instruments are ready in 45 minutes. We don't have any problem with that." (Participant 8)

All dental equipment, instruments, and materials, including those for preventive services, must be made a priority in managers' demand and operational plans. It was found that when the administration is supportive, the working environment is improved and OHP satisfaction is high. This was confirmed by the staff at the referral hospital who reported that they benefit from a supportive working environment. Their equipment was well-maintained and more staff were being recruited to assist in the X-rays section. When dental equipment remains unrepaired for a long period, it shows that the dental services, OHPs, and patients are not valued.

> "Our department has an appointed technician, a medical engineer charged with repairing dental equipment. He is based in the department of maintenance, but when we call him, he immediately comes. He is skilled. Not many things are difficult for him." Participant 13

During the interviews, the OHPs were asked what would ease their work. Firstly, they said they would like the patient/practitioner ratio to be reduced by recruiting more dental staff and

by increasing work stations. This will allow them to work the recommended hours instead of working extra time. Secondly, they would like to have the necessary equipment and materials, a better salary, and tea or coffee breaks at work because even though they like their job, it is hard work. Thirdly, although the organogram of the health centers plans for only one dental therapist, they want it to be revised so that at least two OHPs are hired per facility because the primary healthcare facilities are extremely busy.

## Discussion

This study evaluated oral health services and the challenges faced by OHPs when providing oral healthcare in Rwanda. The study found that there were variations across the seven healthcare centers in terms of human resources, supply chain management, patients education and service rendering, as well as strategic management and administration.

When considering the results of the checklists and the interviews used in the current study, it appears that one of the biggest challenges faced by OHPs while delivering preventive oral health services was linked to human resources. There was a shortage of personnel, such as dentists, specialists and dental assistants. This impinges on oral health service delivery, which is a factor in patient and practitioner satisfaction. This concern has also been raised by researchers in Nigeria and India, where it was found that there is a limited number of dentists compared to the number of people who need dental care [6, 25, 26]. The challenge of understaffing has also been identified in many other low- and middle-income countries, and this limits access to oral healthcare [17].

In addition, most OHPs were dental therapists, and there is a scarcity of dentists and specialist OHPs. The situation is different in South Africa where oral health services are provided by oral hygienists, dental therapists, dentists, and various dental specialists [7]. It was also different from what was reported by a study performed in 47 WHO African countries that found that there are more dentists than dental assistants and dental therapists [5].

Another challenge highlighted in the present study was linked to supply chain management, where it was found that there was a lack of adequate dental equipment, instruments and materials, which was impacting the delivery of preventive dental care [27]. A study performed in Fiji identified the same challenge where OHPs complained of a lack of proper equipment and materials because the available materials were often of poor quality, and this prevented them from performing good-quality work. OHPs in Fiji revealed that they could not offer all services within their scope of practice because they did not have the necessary resources [28]. Similarly, in India, public facilities have been reported to be poorly maintained or to have outdated dental equipment [25]. In the current study, many of the health facilities did not have polishing paste, supposedly because it is not covered by CBHI and patients cannot pay for it themselves. Apart from the polishing paste, periodontal probes were also absent in one health facility and prophylactic prophy angles and prophy cups or brushes were inadequate in general.

However, the results of the current study differ from the results of a study conducted in Tanzania, where only a quarter of facilities had fully functional dental units. OHPs felt that their profession was being neglected [17]. This feeling of being unvalued was shared by the participants in the current study, where most of the OHPs felt that dental services were not valued, and therefore, there were often lags in the procurement process, particularly for preventive equipment and materials. Similar results were reported in Nigeria, where it was found that in the allocation of health resources, decision-makers pay little or no attention to oral health prevention programs because oral healthcare is seen as insignificant compared to other health areas [6]. Similarly, in India, it was found that when primary healthcare systems are

implemented, dental health is not included, leaving dental healthcare far behind other health services in the procurement process [25].

Patients' education and service rendering was another challenge identified in the present study. In fact, participants revealed that it was very challenging and quasi-impossible to provide a full package of oral health education to every patient. They only managed to give postoperative instructions and do some chairside oral health education, especially related to the treatment rendered. Needleman et al. highlights that postoperative oral hygiene instructions may be more important than mechanical plaque removal itself [23, 29]. However, oral health education is vital and it should be comprehensive. Its importance was demonstrated in Honduras, where it was proven that education can help prevent oral diseases because it is key to behavioral change [30]. Communities should be encouraged to adopt healthier lifestyles to live longer and healthier lives [31].

The patient's lack of knowledge and awareness about oral health results in seeking dental care only when they are in pain, and at that time, more invasive and expensive care is necessary. This leads to periodontal problems often being discovered at a late stage when the teeth have already become mobile. In fact, patients worry about their oral health when it becomes painful, and they often visit OHPs with complications from dental caries [26]. The literature shows that early check-ups and regular preventive care reduce the need for dental operative services [32, 33]. Therefore, in the present study, when asked about the mobile dental application, the participants were eager to see a mobile application installed on patients' mobile phones to continuously offer updated oral health information to people, even in rural and remote areas from which most patients come. In fact, it is believed that mobile phones offer a promising strategy for dentists to reach the public and deliver low-cost oral health education and promotion. A mobile application for oral health education has been proven to be effective in teaching patients how to take care of their oral health at their convenience time and place [34]. The application can be used to teach the population how to maintain good oral health, proper diet and nutrition, and the importance of routine dental check-ups [35]. If people know how to prevent oral diseases, there will be less dental patients, and patient/practitioner ratios will improve [33]. The importance of oral health promotion and education has been also highlighted in other studies. Interventional studies performed in different countries demonstrated that the attitude of parents who were taught about the oral health of their children in the intervention group improved and contributed to a significant increase in the oral health status and oral hygiene of their children, while in the control group, children had more dental caries on their deciduous molars [36–38]. This confirms that strong oral health promotion and education is the basis for oral health solutions.

In the present study, the findings showed that another challenge faced by OHPs is linked to strategic management and administration. In fact, it is the responsibility of health facility administration to ensure the recruitment of enough staff and the procurement of dental equipment, instruments and materials. However, even the few dental personnel who are available feel demotivated. Even though OHPs reported that they love their profession, they could not deny that it is tiresome, and therefore, they would appreciate and be more motivated if they received financial bonuses and incentives such as tea breaks during work hours. Suga et al. found that dentists are willing to provide dental caries preventive measures if their payment schemes were fairer [33]. The management of public health facilities should review their planning system so that dental services are equipped with essential equipment, basic dental materials and instruments, and proper human resources.

The management should also advocate for patients in insurance companies or provide standard required treatment regardless of the insurance cover. In this study, it was discovered that the fact that dental polishing is not covered by CBHI discourages dental practitioners from

providing dental polishing treatment, which hasten the dental plaque accumulation on scaled dental surfaces. The impact of medical insurance on dental service accessibility has also been raised by Ramraj et al. and Uguru et al. because they found that not only low-income groups but also middle-income groups struggle to access comprehensive dental care because some acts are not covered by medical insurance [6, 39]. Therefore, it is not enough to have health insurance, and the insurance should also cover at least all basic dental services, such as dental polishing [40, 41]. The problem of health insurance covering only a few dental procedures was also found in Nigeria, where some treatments are not provided due to lack of insurance coverage [6]. The CBHI should be encouraged to cover polishing because it is not very expensive. Basic oral healthcare coverage, including prevention, is patients' right. Efforts must be made by the management to support and advocate for preventive oral health service coverage, such as polishing, through the CBHI. This is highlighted by Bastani et al., who emphasized that policymakers in low- and middle-income countries should aim to improve populations' access to oral and dental services through advocacy for comprehensive insurance packages because millions of people in developing countries cannot afford basic dental treatments [42]. Hayashi et al. said that dentistry should move from an increasingly unaffordable curative model to a cost-effective evidence-based preventive model [31].

## Limitations

The limitations of this study were that participants were recruited from a single district and only from government institutions, which means the results cannot be generalized to the entire country or private facilities. The quantitative data that was measured using an observation checklist was not all-inclusive, and some of the equipment/consumables required (e.g: Restorative materials, airotors) were not included in the audit. The study checklist focused on preventive oral hygiene services and required materials and equipment used for scaling and polishing.

## Conclusion

The challenges faced by OHPs in Nyarugenge District were related to the following areas: human resources, supply chain management, patients' oral health awareness and service rendering, and strategic management and administration. However, if one looks closely, it appears that all these challenges can be addressed by proper management of the health facilities. Proper and optimal strategic management of public health facilities would make an effort to recruit enough personnel which would free time for OHPs to provide oral health education and other preventive measures and to do comprehensive oral examinations, which would allow early detection and, thus prevention, of oral diseases [43]. Furthermore, the management of health facilities should ensure the availability of dental equipment, instruments and materials for oral disease prevention and remember to prioritize community oral health promotion and education in order to raise the awareness of the population towards oral health. Well-established community preventive interventions such as a dental mobile application would help improve patient/provider ratios because it would raise awareness about oral health and help people adopt healthy behaviors. Finally, the management of health facilities should also make advocacy for the population so that more dental services would be covered by health insurance, especially the CBHI, which would allow OHPs to offer oral health services without constraints.

## Supporting information

**S1 Dataset.**
(ZIP)

## Acknowledgments

We appreciate the biostatisticians at the University of the Witwatersrand and the CARTA facilitators for their support during the initial data analysis.

## Author Contributions

**Conceptualization:** Emmanuel Nzabonimana, Yolanda Malele-Kolisa, Phumzile Hlongwa.

**Data curation:** Emmanuel Nzabonimana, Yolanda Malele-Kolisa, Phumzile Hlongwa.

**Formal analysis:** Emmanuel Nzabonimana, Yolanda Malele-Kolisa, Phumzile Hlongwa.

**Funding acquisition:** Emmanuel Nzabonimana.

**Investigation:** Emmanuel Nzabonimana, Yolanda Malele-Kolisa, Phumzile Hlongwa.

**Methodology:** Emmanuel Nzabonimana, Yolanda Malele-Kolisa, Phumzile Hlongwa.

**Project administration:** Emmanuel Nzabonimana, Yolanda Malele-Kolisa, Phumzile Hlongwa.

**Resources:** Emmanuel Nzabonimana, Yolanda Malele-Kolisa, Phumzile Hlongwa.

**Software:** Emmanuel Nzabonimana, Yolanda Malele-Kolisa, Phumzile Hlongwa.

**Supervision:** Yolanda Malele-Kolisa, Phumzile Hlongwa.

**Validation:** Emmanuel Nzabonimana, Yolanda Malele-Kolisa, Phumzile Hlongwa.

**Visualization:** Emmanuel Nzabonimana, Yolanda Malele-Kolisa, Phumzile Hlongwa.

**Writing – original draft:** Emmanuel Nzabonimana, Yolanda Malele-Kolisa, Phumzile Hlongwa.

**Writing – review & editing:** Emmanuel Nzabonimana, Yolanda Malele-Kolisa, Phumzile Hlongwa.

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
