## [Decision Letter · Decision Letter 0]

24 Jul 2024

PONE-D-23-33199Evaluation of oral health services and challenges facing oral health practitioners working in Nyarugenge, RwandaPLOS ONE

Dear Dr. Nzabonimana,

Thank you for submitting your manuscript to PLOS ONE. After careful consideration, we feel that it has merit but does not fully meet PLOS ONE’s publication criteria as it currently stands. Therefore, we invite you to submit a revised version of the manuscript that addresses the points raised during the review process.

We look forward to receiving your revised manuscript.

Kind regards,

Ana Cristina Mafla

Academic Editor

PLOS ONE

Journal Requirements:

"“EN was supported by the Consortium for Advanced Research Training in Africa (CARTA). CARTA is jointly led by the African Population and Health Research Center and the University of the Witwatersrand and funded by the Carnegie Corporation of New York (Grant No. G-19-57145), Sida (Grant No:54100113), Uppsala Monitoring Center, Norwegian Agency for Development Cooperation (Norad), and by the Wellcome Trust [reference no. 107768/Z/15/Z] and the UK Foreign, Commonwealth & Development Office, with support from the Developing Excellence in Leadership, Training and Science in Africa (DELTAS Africa) programme. The statements made and views expressed are solely the responsibility of the Fellow”"

4. In the online submission form, you indicated that "Data are available upon reasonable request from the corresponding authors."

**Additional Editor Comments:**

Dear authors,

I found the article interesting and important for the community. However, there are some details that need to be improved:

1. Check the number of questions you used in the interviews, you mentioned 13 but only 7 are described.

2. Mention the place of manufacturing of the sotfwares used in the study.

3. Use only one number after the point when you use percentages, otherwise do not use numbers after the point in percentages (there is not any precision here because of small frequencies).

4. Table 1, only should use frequencies, and probably summary a Total column with a frequency and percentage.

5. The Tables are not very useful in this case, evaluate what tables really you need, you could summarize the information in a paragraph.

6. Remove the box of the Discussion section.

Reviewers' comments:

Reviewer's Responses to Questions

**Comments to the Author**

1. Is the manuscript technically sound, and do the data support the conclusions?

Reviewer #1: Yes

2. Has the statistical analysis been performed appropriately and rigorously? 

Reviewer #1: No

3. Have the authors made all data underlying the findings in their manuscript fully available?

Reviewer #1: Yes

4. Is the manuscript presented in an intelligible fashion and written in standard English?

Reviewer #1: Yes

5. Review Comments to the Author

Reviewer #1: This study is interesting and offer essential information. However, there are a few concerns:

1. Please check the grammar throughout the manuscript.

In Title: (for example, it has to be '..challenges faced by..' instead of '..challenges facing..'

In introduction: '... This means public institutions are full of poor people...' Rephrase such sentences.

2. Why is the theme and description table repeated again in DISCUSSION section?

3. Is the quantitative data that is measured using an observation checklist all-inclusive? The authors have not made a note / mentioned about the other necessary equipment / consumables required (eg: Restorative materials, airotors, etc.)

6. PLOS authors have the option to publish the peer review history of their article (what does this mean?). If published, this will include your full peer review and any attached files.

Reviewer #1: No

---

## [Author Response · Author response to Decision Letter 0]

2 Aug 2024

Response to reviewers attached as file.

---

## [Editor Report · Decision Letter 1]

7 Aug 2024

Evaluation of oral health services and challenges faced by oral health practitioners working in Nyarugenge, Rwanda

PONE-D-23-33199R1

Dear Dr. Emmanuel Nzabonimana,

We’re pleased to inform you that your manuscript has been judged scientifically suitable for publication and will be formally accepted for publication once it meets all outstanding technical requirements.

Kind regards,

Ana Cristina Mafla

Academic Editor

PLOS ONE

---

## [Editor Report · Acceptance letter]

9 Aug 2024

PONE-D-23-33199R1 

PLOS ONE

Dear Dr. Nzabonimana, 

I'm pleased to inform you that your manuscript has been deemed suitable for publication in PLOS ONE. Congratulations! Your manuscript is now being handed over to our production team.

Kind regards, 

on behalf of

Dr. Ana Cristina Mafla 

Academic Editor

PLOS ONE